# "A sense of the bigger picture:" A qualitative analysis of follow-up interviews with people with bipolar disorder who self-reported psilocybin use

**Meghan DellaCrosse**[1,2,3,4]*, **Mollie Pleet**[2,3], **Emma Morton**[5], **Amir Ashtari**[5], **Kimberly Sakai**[2,3], **Josh Woolley**[2,3‡], **Erin Michalak**[5‡]

1 Department of Clinical Psychology, The Wright Institute, Berkeley, California, United States of America, 2 Department of Psychiatry and Behavioral Sciences, University of California, San Francisco, San Francisco, California, United States of America, 3 San Francisco Veteran's Affairs Medical Center, San Francisco, California, United States of America, 4 Department of Psychiatry and Behavioral Health, The Ohio State University, Columbus, Ohio, United States of America, 5 Department of Psychiatry, University of British Columbia, Vancouver, British Columbia, Canada

‡ JW and EM co-last authorship on this work.
* mdellacrosse@wi.edu

**Data Availability Statement:** Available data relevant to this study is presented here in the

## Abstract

### Objectives

People with bipolar disorder (BD) spend more time depressed than manic/hypomanic, and depression is associated with greater impairments in psychosocial functioning and quality of life than mania/hypomania. Emerging evidence suggests psilocybin, the psychoactive compound in "magic mushrooms," is a promising treatment for unipolar depression. Clinical trials of psilocybin therapy have excluded people with BD as a precaution against possible adverse effects (e.g., mania). Our study centered the experiences of adults living with BD who consumed psilocybin-containing mushrooms, and aimed to (1) understand its subjective impacts on BD symptoms, (2) deepen understanding of Phase I survey results, and (3) elucidate specific contextual factors associated with adverse reactions in naturalistic settings.

### Methods

Following an international survey (Phase I), follow-up interviews were conducted with 15 respondents (Phase II) to further understand psilocybin use among adults with BD. As part of a larger mixed-methods explanatory sequential design study, reflexive thematic analysis was used to elaborate findings.

### Results

Three major themes containing sub-themes were developed. (1) Mental Health Improvements: (1.1) decreased impact and severity of depression, (1.2) increased emotion processing, (1.3) development of new perspectives, and (1.4) greater relaxation and sleep.

manuscript text and Supporting information. This study is protected by a NIH Certificate of Confidentiality. Given the content on illegal behavior and the detailed narrative structure of each interview, whole transcripts cannot be completely de-identified to be shared publicly.

**Funding:** The authors received no specific funding for this study.

**Competing interests:** EEM has received support for patient educational events from Otsuka-Lundbeck Foundation. JW is currently consulting on scientific protocol development for Alvarius Pharmaceuticals, and previously consulted on protocol development for Psilo Scientific Ltd. and Silo Pharma. This does not in any way alter our adherence to PLOS publishing policies.

(2) Undesired Mental Health Impacts: (2.1) changes in sleep, (2.2) increased mania severity, (2.3) hospitalization, and (2.4) distressing sensory experiences. (3) Salient Contextual Factors for psilocybin use included: (3.1) poly-substance use and psilocybin dose, (3.2) solo versus social experiences, and (3.3) pre-psilocybin sleep deprivation.

## Conclusion

Our findings demonstrate both benefits and risks of psilocybin use in this population. Carefully designed clinical trials focused on safety and preliminary efficacy are warranted.

## Introduction

### Bipolar disorder

Impacting approximately 2.4% of the global population, bipolar disorder (BD) is among the leading causes of disability worldwide [1]. Individuals with BD experience on average three times more days depressed than manic or hypomanic [2], and episodic depression is the most frequent cause of disability among individuals with the various subtypes of BD [3]. In people with BD, depressive episodes and residual symptoms account for more long-term morbidity, impairment in psychosocial functioning and quality of life, and risk of suicide than manic or hypomanic episodes [4–9].

Currently available pharmacological treatments for BD-associated depression are inadequate with inconsistent efficacy, significant side effects, and the risk of possibly precipitating "affective switching" [10, 11]. The cycling nature of BD poses particular challenges to clinicians and researchers [12] and controversy and lack of consensus continues regarding the use of serotonergic antidepressants for BD depression [10, 13–15]. While some have advised against it entirely [13], others have questioned the perceived causality of antidepressants and emergent hypo/mania given affective changes associated with this condition [10]. Given the clinical need and limited established treatments for depression in BD, new treatment approaches are desperately needed [12, 16].

### Recent resurgence in psychedelic research

Psilocybin is a naturally occurring compound found in a variety of mushroom species (sometimes referred to as "magic mushrooms") [17]. When consumed, psilocybin is metabolized into *psilocin* [18, 19], leading to dose-dependent psychedelic effects through psilocin's agonist action at serotonin receptors (5-HT) (mostly 2A but also 1A, and 2C) [20]. These mushrooms have been used in ritual and healing contexts for thousands of years [21, 22], and despite schedule I status designation [23], an estimated 9.68% of adults in the United States have used psilocybin at least once [24].

Growing evidence suggests psilocybin can be an effective treatment for unipolar depression [25–28]. These findings led the Food and Drug Administration (FDA) to designate psilocybin as a "breakthrough therapy" for depression [29, 30]. This designation demonstrates the FDA's acknowledgment that the drug may outperform other available therapies, and thus expedites the Federal review of psilocybin. However, despite overlaps between unipolar depression and depression in the context of BD, modern clinical trials universally exclude individuals with any BD diagnosis or a family history of BD [19, 31]. Little justification is typically provided for this and there is a dearth of published literature on the safety of psychedelic use in people with BD.

One major reason for exclusion is the belief among clinicians and researchers that the serotonergic action of psilocin could precipitate Treatment Emergent Affective Switching (TEAS) or some other adverse experiences [32] similar to antidepressant monotherapy for this population [10, 11, 33, 34]. TEAS characterizes what is typically a switch from a depressive episode to hypo/mania [33, 34]. Whether this pattern of emergent hypo/mania following depression is due to antidepressant treatment is yet to be fully understood [12, 34, 35]. Furthermore, it remains unclear to what extent the non-static nature of depression in this population contributes to these unresolved dilemmas and thus treatment options [36]. People with BD have been excluded from psychedelic research [19, 31] despite a lack of clarity regarding the cause or mechanism of TEAS. This has contributed to a lack of clear evidence that psilocybin actually induces TEAS and if so, to what degree.

At present, information on psilocybin use among people with BD is limited to case reports. In a review of published case reports, 17 individuals were found with a clear description of emergent BD symptoms after psychedelic use (i.e., psilocybin, ayahuasca, LSD, DMT) [32]. Across the cases, emergent manic symptoms or adverse reactions were associated with factors such as poly-substance use, repeated drug consumption, and large doses. Out of 17 cases, two described people with a history BD who experienced symptoms of mania following psychedelic use (not psilocybin though). However, one of these individuals used the psychedelic (vaporized DMT) daily for six months [37]. The other case reported on an individual who was experiencing hypomania two weeks prior to psychedelic use in the context of a 4-day ayahuasca retreat [38]. Four of the cases described people without a diagnosis of BD, although two had family histories, who took a single psychedelic substance and developed sustained manic symptoms [39–42] without any other precipitating factors like poly-substance use. The only one of these cases that involved psilocybin was a 21-year-old woman with a family history of BD who initially had a pleasant experience during her first psilocybin use. However, 36 hours later she was hospitalized for reduced sleep and symptoms of mania with psychotic features. She was stabilized using lithium and aripiprazole, and later lamotrigine [42]. The authors concluded that development of mania following psychedelic use can occur, but the rates are likely low given the widespread use of these compounds and the paucity of cases that did not involve polysubstance use or other clear instigating circumstances [32]. Given the variability and other limitations inherent to case studies, in addition to family history exclusionary criteria, more research is needed to further understand the risks for this population.

## Current study

The over-arching mixed-methods study from which the present data is drawn was designed to elicit understanding of psilocybin use among adults with BD, including use patterns, possible benefits and adverse event profiles associated with consumption, and contextual factors framing use behavior to provide recommendations for future clinical trials. In Phase I, Morton and colleagues [43] used an international web-based survey to explore experiences of individuals with self-reported BD diagnoses who had experience using psilocybin mushrooms. Of 541 completed responses (46.4% woman-identifying, mean age 34.1 years old), one-third (32.2%; $n = 174$) of respondents reported new or increasing symptoms following psilocybin (e.g., mania, sleep challenges, anxiety). However, use of emergency medical services was rare (n = 18; 3.3%), with psilocybin experiences typically reported as more helpful than harmful. The current qualitative Phase II study centers the experiences of adults living with BD who consumed psilocybin mushrooms and aims to understand the subjective impact of this psychedelic on BD symptoms. Qualitative approaches offer important perspectives complimentary to quantitative means [44] to gain a comprehensive understanding of a problem [45], and allow

for inquiry on emergent topics such as psychedelic science where theoretical models remain nascent [46]. A subsample of Phase I survey respondents were invited to participate in follow-up interviews designed to deepen understanding of the preliminary survey results, and elucidate specific contextual factors associated with adverse reactions in naturalistic settings. Considering the variability of lived experience and treatment options for individuals with bipolar depression [12], a qualitative phase of the study is crucial to understanding the benefits, risks, and contextual factors that will inform future research in the field of psychedelic science.

## Methods

### Study design

Utilizing an explanatory sequential design [47, 48] this study was conducted by collaborators from the Translational Psychedelic Research Program (TrPR) at the University of California, San Francisco (UCSF), and the Collaborative RESearch Team to study psychosocial issues in Bipolar Disorder (CREST.BD) at the University of British Columbia (UBC). The study received approval from the UCSF Independent Review Board (IRB#20–31330).

Data collection for Phase I of the broader study took place from October 2020 through January 2021 in the form of an international self-report survey, results of which are published elsewhere [43]. Qualitative findings in this present analysis were collected through semi-structured interviews of consenting survey respondents recruited from Phase I. The aim of follow-up interviews was to elaborate on the experiences of individuals with BD who reported use of psilocybin to achieve a psychedelic experience. Interviews took place on a rolling basis between December 2020 and May 2021.

### Recruitment

**Inclusion/exclusion criteria.** Participants recruited to Phase I of this study were required to meet the following initial criteria: (1) at least 18 years of age, (2) have a self-reported diagnosis of BD (any subtype), and (3) have self-reported prior experience of taking at least one full dose of psilocybin mushrooms. For Phase II, participants were required to: (1) reside in North America, (2) report willingness to be contacted for follow-up interview, and (3) meet DSM-5 criteria for a BD diagnosis as confirmed by a semi-structured clinical interview. No restrictions were placed on current symptom severity. To maximize the diversity of perspectives, researchers sought a purposive subsample [49] of survey respondents with varying demographic details (i.e., race/ethnicity, gender identity, geographical location) and reported outcomes of mushroom use. Particular effort was made to include participants who endorsed (in Phase I) extremely positive or extremely negative experiences with psilocybin mushrooms. Individuals with experience limited to 'microdosing' were excluded as such doses are said to be subperceptual [50].

### Qualitative interview procedures

**Informed consent.** Participants were consented to participate in this study by the designated qualitative interviewer (MP) prior to any study procedures. Participants were informed their interviews would remain confidential, and their transcripts would be de-identified and stored on encrypted and HIPAA compliant UCSF servers. Participants were given the option to request follow-up contact from the research group when resulting research is disseminated.

**Diagnosis.** Prior to interview, participants were administered a Structured Clinical Interview for DSM-5 (SCID) to confirm diagnosis. Inclusion criteria required a confirmatory diagnosis of Bipolar Type I, Bipolar Type II, Cyclothymia, or BD Not Otherwise Specified.

Assessment interviews were conducted by Masters level clinical assessors from the UCSF TrPR Program over a secure connection via Zoom.

**Data collection.** Qualitative data was collected via semi-structured interviews conducted via Zoom lasting 1–1.5 hour and were audio-video recorded. Participants were reminded of recording procedures and were given the opportunity to ask questions prior to commencing the recorded interview. An interview guide was used, consisting of specific questions centered on participant psychedelic experience(s) following psilocybin mushroom use.

**Interviewer.** All interviews were conducted by a single clinician-researcher member of the TrPR Program (MP) who had received extensive training and support from expert qualitative researchers of the UCSF and UBC joint study team.

**Semi-structured interview guide.** The multi-item interview guide (see S1 File) was developed collaboratively among study team members from the TrPR Program and CREST.BD, as well as individuals living with BD on the CREST.BD Community Advisory Group. This guide included a range of questions designed to address four overarching questions of therapeutic interest: (1) Why do people with BD use psilocybin? (2) In which contexts and situations do people with BD use psilocybin? (3) What occurs when individuals with BD use psilocybin? (4) What intentions do individuals with BD hold regarding future psilocybin use.

## Analysis: Coding and thematic analysis

Audio-video recorded Phase II follow-up interviews were transcribed by the HIPAA compliant company Home Row and uploaded onto the qualitative data analysis software NVivo (QSR International Pty Ltd., Doncaster, AU). A preliminary coding framework was developed by researchers MD, MP, EM and EEM; group members independently coded two text samples selected from interview transcripts followed by collaborative review. Researchers MD and MP next coded three transcripts independently before meeting to establish a shared understanding of meaning [51]. MD and MP met at regular intervals to engage in the self-reflexive and iterative process of thematic analysis by (1) becoming familiar with the findings, (2) generating initial codes, (3) searching for themes, (4) reviewing themes, (5) defining themes, and (6) writing-up the results [52, 53]. MD and MP engaged in a continual familiarization process with the data that involved reading and re-reading transcripts individually, meeting weekly for discussion, and meeting monthly for review with the broader team.

Reflexive thematic analysis (TA) as described by Braun and Clark [51–56] as well as Terry and colleagues [57] was selected for several reasons. First, TA offers a flexible approach to theme development across cases [51, 58] relevant to an understudied topic and population. Second, the recursive process of TA coding and theme development, including engagement and deep reflection on the findings, was seen as supportive of research goals to further understand the subjective experiences of the study population. The important "double consciousness" or analytic eye described by Braun and Clark [54] calls upon the researcher to listen intently while critically reflecting on what is being expressed.

## Lived experience expertise and engagement

Phases I and II of the study were conducted within the context of a Community-Based Participatory Research (CBPR) model. CBPR involves actively including community members in the research process to increase contextual understanding and deliver equitable benefits to the community [59]. One co-author on this paper has lived experience with both BD and psilocybin use; in Phase I they contributed to study design and data analysis, and in Phase II to study design, interpretation of findings and manuscript development. Survey questions in Phase I were developed collaboratively by team members from the TrPR Program and the

Collaborative RESearch Team to study psychosocial issues in BD (CREST.BD), with input from individuals living with BD from the CREST.BD Community Advisory Group.

## Results

Of the 332 survey participants who indicated willingness to be contacted for follow-up interview, (283 from USA; 49 from Canada), 23 were invited and 15 individuals were interviewed (2 people declined to participate and 6 were non-responsive to the invitation). Participant ages ranged between 27–50 years, and the majority of participants' self-identified gender was woman (9 women; 5 men; 1 genderqueer). Following SCID diagnosis, 13 participants were given a diagnosis of BD-I, and 2 participants were diagnosed BD-II (see Table 1).

Here, we focus on reporting themes related to psilocybin use outcomes, namely: (1) mental health improvements, (2) undesired mental health impacts, and (3) contextual factors of beneficial and adverse psilocybin experiences. Sub-themes representative of variations and distinctions within the broader themes are reported below. Narrative excerpts are provided to illustrate these themes and sub-themes from the participants' perspectives.

### Mental health improvements

Participant reports of mental health improvements included experiences of (1) decreased impact and severity of depression, (2) increased emotion processing, (3) development of a new perspective following psilocybin consumption, and (4) greater relaxation and sleep. Participants remarked on how their psilocybin experiences enabled emotion processing and development of new perspectives that supported positive changes in mood. Experiences of greater relaxation were described in terms of enhanced calmness and grounding that in some cases contributed to improved sleep.

**Table 1. Demographic data.**

| Participant number | Diagnosis (from SCID) | Self-identified gender identity (Survey: three choices or self-described or not to answer) | Age at interview | Racial/ethnic identity (Survey: self-described) |
|---|---|---|---|---|
| 101 | BD1, w/ psychotic features | Man | 32 | White |
| 103 | BD1 | Woman | 46 | Black |
| 106 | BD1 | Man | 36 | Lebanese Canadian |
| 108 | BD1 | Woman | 50 | Caucasian Nigerian |
| 109 | BD1 | Woman | 34 | Latina |
| 111 | BD1 | Man | 40 | Latino, Black |
| 112 | BD1, w/ psychotic features | Woman | 34 | Canadian European |
| 113 | BD1 | Woman | 31 | Vietnamese |
| 114 | BD2 | Queer | 30 | Biracial (Black, White, Native Latino) |
| 115 | BD1 | Woman | 35 | White, Jewish |
| 117 | BD1 | Woman | 27 | White |
| 120 | BD1 | Woman | 39 | White |
| 121 | BD1, w/ psychotic features | Man | 30 | Caucasian |
| 122 | BD2 | Man | 29 | Mexican |
| 123 | BD1 | Woman | 39 | Caucasian |

**Decreased impact and severity of depression.** Mental health improvements were demonstrated in some participants through accounts of decreased symptom severity and frequency of depressive episodes following psilocybin consumption, both over the short-, and longer-term.

In speaking about their subjectively experienced improvements in mental health, 123-BD1 noted:

"I felt like it lifted my mood. Not just like mood, but like what I had been feeling like that month, you know."

Even when participants reported continued depressive symptoms in the weeks or months after psilocybin use, many remarked that the severity of the depression remained improved, such as 122-BD2:

"[. . .] you know, I'll still feel the depression. I'll still feel like super anxious I want to like crawl out of my skin. But for the most part, like I'm still able to get up and do what I still need to get done."

Many participants noticed these changes in depression fairly quickly. For instance, as 103-BD1 shared, "In the time fairly close after [using psilocybin] I noticed a much lowering of my anxiety, and also my depression." However, symptom improvements were commonly reported to extend longer-term. Participant 103-BD1 said:

"Overall, since [taking psilocybin]—I mean, because that was in 2009—I have never had as bad of depression since that period. It may be coincidence, may not, I don't know, but I do know that I have never had a depression that was [. . .] incapacitating."

In addition to naming alleviation of depression and associated impairment as a primary benefit, some participants processed the impact of improved mood on their sense of self-efficacy. Participant 112-BD1 said:

"So, afterwards I feel a lot more capable in my life because, like, depressive symptoms can be so debilitating and feel, like, really helpless and, like, powerless."

**Increased emotion processing.** A pattern of increased emotion processing, i.e., novel exploration of inner experiences (e.g., emotion, memories) in which participants described greater openness to emotional experience, was observed in several reported experiences during psilocybin use. Participants reflected on this emotion processing and reported that psilocybin facilitated "going inward" and "not having to police [their] feelings" (114-BD2). Participants viewed their ability to contact and then work through suppressed emotion, such as grief, as a catalyst for helping lift depressive symptoms. Additionally, therapeutic insight and cathartic release were noted by some to be a part of this emotion processing.

For instance, 113-BD1 stated:

"I would say that [psilocybin] helped me process [my emotions] better [. . .] I took 2.5 grams, I threw up, and I ended up crying a lot. So, I ended up crying but, like, in a grieving way. Like, a way that, like, I felt that, like, I was mourning my child self or my previous self."

Participant 115-BD1 remarked on her experiences of psilocybin-supported inner exploration:

"I can just, like, go lie down and open up to [my emotions] and explore [. . .] [the psilocybin experience] is more internally guided [than psych medication]."

By allowing for emotional release, psilocybin was described by some as a means to enable greater ease in functioning. Using metaphoric imagery to detail this process, participant 112-BD1 noted:

"[. . .] It's almost like [. . .] those movies where there's, like, vines, like, holding somebody back and, like, psilocybin is, like, you know, the machete that breaks through and, like you're able to, like, kind of get that off and, like, be able to move again."

**A new perspective.** For some participants, psilocybin use offered new perspectives, enabling a more positive outlook of themselves (including their experience of BD symptoms), which was said to benefit their mental health.

As 122-BD2 illustrated, "I use psilocybin as more of a tool to like kind of help and guide. And like, you know, offer new perspectives and, you know, most importantly, help for like the depression aspect of my life." When psilocybin was taken to address low mood, some participants shared that these new perspectives served to protect them against drifting back into a depressive mindset. Participant 108-BD1 exemplified this in saying:

"With depression, I notice that I'm not as concerned because I feel like I have a bigger picture. Like, I have the bigger picture when I experience the opening that magic mushrooms offer. So, the things that can bring me into a depression, even if it's just the kind of undercurrent of blah, doesn't seem as potent, I guess."

This sense of the "bigger picture" was noted to further protect some participants against depression by distancing them from the daily challenges and offenses that would otherwise lead their moods to spiral. Participant 109-BD1 explained:

"[. . .] in the days after and sometimes up to weeks I just feel kind of, like, more protected in a way and, like, it feels like things can kind of, like, roll off my shoulders easier. They don't impact me as much."

Although peak psilocybin effects were regularly credited for these positive shifts in perspective, the days and weeks following acute effects were also noted intrinsic to the therapeutic gain. Participant 112-BD1 shared that the timeframe after consuming psilocybin mushrooms supported her in synthesizing this "new perspective," which allowed her to engage in life with more enthusiasm:

"And I feel more motivated for life [. . .] Like, it's exciting, the integration process, I think, is what's really exciting about kind of taking, like, that new perspective, some of the knowledge, and kind of seeing how it fits into my life moving forward."

**Greater relaxation and sleep.** Despite a sizeable portion of participants who reported undesired mental health impacts in relation to sleep changes (see *Changes in sleep*), some

participants cited greater relaxation as a positive outcome of their psilocybin experience. In some instances, this increased calmness reportedly benefitted their sleep. Participant 109-BD1 described this feeling as "calmness that kind of comes over me," and said, "it's not that anything externally has shifted, it's more that I feel more capable of managing what's happening in my world at that given moment." Participant 103-BD1 remarked that her experience of relaxation was surprising:

"I was shocked. I'm just like, oh, yeah, okay. And that was definitely not my normal mode. I'm not that relaxed."

In another account, 113-BD1 remarked on the relaxing effect and said that "psilocybin helps [her] be grounded." She went on to explain:

"I feel like it's clear, I'm calm, I'm grounded. Which I was so amazed because I didn't know I could feel that way. Like, I remember the first time taking it, and even after taking it, like, I felt grounded. I felt heavy in my body and it felt so cool."

Participant 103-BD1, who said she's "always had trouble falling and staying asleep," spoke positively about the sense of relaxation during her psilocybin experience, which she attributed to her subsequent sleep improvement:

"I slept a bit more. Like, I didn't oversleep like the depression sleep [. . .] I was able to relax. And we often fell asleep to music, so it's like, get the music on, you're relaxed, next thing you know it's morning, I've slept all night, I didn't get up in the middle of the night, and it was great."

Participant 106-BD1 said, "sometimes you eat [mushrooms] and they make you really sleepy [. . .] it's happened to me a few times." Similarly, participant 111-BD1 remarked on how his partner observed him sleeping more and earlier in the couple of weeks following his psilocybin experience:

"She kind of thought, *that's weird* because you don't normally sleep that early in the day. And that was probably [. . .] in the two weeks after I did the mushrooms."

### Undesired mental health impacts

Participant narratives of undesired mental health impacts included (1) changes in sleep, (2) increased mania severity, (3) hospitalization, and (4) distressing sensory experiences. Sleep changes following psilocybin use were prominent among participant reports. Difficulty relaxing or sleeping after their drug experiences led some participants to experience worsening of symptoms, which at times presented as hypo/mania or distressing sensory experiences that led some participants to become hospitalized.

**Changes in sleep.**  While some reported no changes or even improved sleep (see *Greater relaxation and sleep*), changes in sleep were prominent among participant reports of undesired mental health impacts following psilocybin use. For some, these problems involved difficulty relaxing enough to fall asleep, as well as poor sleep quality and agitation.

In a description of her experience following psilocybin use, participant 123-BD1 said, "sometimes, I think [. . .] my mind was still kind of going, you know." To further illustrate the issue, 120-BD1 shared, "I just didn't sleep until [the psilocybin] was like completely out of

my system." A few participants stated that insomnia could last significant periods of time. For example, 106-BD1 explained:

> "It's hard to go to sleep after mushrooms [. . .] a lot of times you'll be up all night and stay up the whole next day."

Participant 112-BD1 shared that even though she may be physically tired after a day of psilocybin use, nighttime is often when she becomes mentally alert and creative. She reported experiencing urges to think, draw, journal, or engage in other creative or introspective activities throughout the night. She said:

> "I can't fall asleep until, like, usually the next afternoon. That's usually around the time. And at that point, like, I have that headache and I am super tired. And sometimes, like, I'll be lying in bed and, like, I'm so tired but, like, my mind just, like, won't, like, just fall asleep. Which is—that's not comfortable, it just sucks."

When some participants were able to fall asleep the night following psilocybin use, their sleep quality was often described as poor, or in 121-BD1's experience, "a bit agitated." For instance, 123-BD1 shared that after using psilocybin mushrooms, she remembers having "a bit more wakeups in the night." In describing post-psilocybin sleep issues, 121-BD1 stated:

> "It was not a very qualitative sleep. It was a bit too agitated. I was not feeling very rested when I woke up in the morning. But I slept for about seven, eight hours."

To address post-psilocybin sleep issues, some participants reported using quiet activities such as reading, over the counter medication (e.g., Nyquil, Benadryl), as well as prescription medications. For instance, after a long psilocybin experience that entailed disturbing auditory hallucinations, 115-BD1 used pharmaceutical medications to prevent the seemingly inevitable insomnia that would otherwise follow. She stated:

> "I don't know what time I eventually went to bed, but I just was like [. . .] I can't sleep. Like [. . .] I'm just not going to sleep. So, I think I eventually ended up taking some maybe, like, extra Seroquel and maybe some Klonopin or something. But that night, I mean, I can just be up until 5 [am] or whatever if I don't take something."

**Increased mania severity.** Multiple participants remarked on how their experience of psilocybin seemed to precipitate hypomanic or manic episodes. As 112-BD1 shared:

> "The only time I go into, like, a full-blown manic episode is when I've done psychedelics. It, like, unlocks it for me."

In describing the relationship between psilocybin and mental health, participant 112-BD1 reported getting a "psychedelic hangover" following psilocybin use that she experienced as a sort of mania. In describing this experience, 112-BD1 stated:

> "[The 'hangover'] can last anywhere—I've had ones that it was, like, three to four days, and that's, like, a bit—like, it's manic but not, like, full-blown, like, I feel out of control manic. And I've had one that lasted a week."

Although not a commonly reported experience, 103-BD1 shared that since using psilocybin mushrooms, her experience of manic symptoms has transitioned into a more frequent but less intense hypomania. She said:

"Before the mushrooms, I had a lot more depressive episodes, and I'd had a couple of full-blown manic episodes. But, after that, I haven't had a full-blown manic episode since then. I have been hypomanic many times [. . .]."

For some, the relationship between psilocybin and [hypo]/mania appears in a fixed pattern. However, other participants noted an inconsistent relationship between these two states. After reflecting on her experience of having a post-psilocybin manic episode, 108-BD1 reported:

"I've had times where I've taken the same sort of [mushroom] dose and I've had no spike. So, my belief is that [psilocybin] has no effect on the manic symptoms."

**Hospitalization.**   Psychiatric hospitalization was among the most severe outcome that participants reported in relation to psilocybin use. Worsening symptoms that led to this event included mania, psychotic symptoms, grandiosity, and impaired decision-making. For instance, 106-BD1 said:

"In the last year, I haven't eaten [mushrooms] as much because [. . .] I had a full manic episode that happened back in 2018. So, from there I got hospitalized and then I couldn't eat them as often as I wanted to."

Participant 106-BD1 described the experience precipitating hospitalization by sharing that after "medicating" with psilocybin mushrooms for "24 hours," he had a strong spiritual epiphany:

"I woke up from that trip and I was just like, I'm Christ. I just thought I was Christ. And at first, I thought I was Jesus. I thought I was the reincarnation of Jesus [. . .] And then I figured out that, no, you're not the reincarnation of Jesus, you're just full of Christ energy."

Participant 108-BD1 reflected on her experience of being hospitalized after neighbors reported her for exhibiting unusual behavior. She stated:

"This experience of being hospitalized was related to two things, the grandiosity, feelings of being able to do something about what I'm witnessing in the world [. . .] And one of the neighbors called the police and said that I've never behaved in such a manner before."

Participant 115-BD1 endorsed using psilocybin mushrooms multiple times within a 3-month period before being hospitalized for a manic episode with mixed features. Although she reported uncertainty as to whether the mushrooms were implicated in her hospitalization, she expressed openness to this being possible:

"I went into the hospital for [the mixed manic episode], and I—one thought that I had, which is a possibility, is, like, [. . .] I had tried [consuming mushrooms] maybe, like, four or five times, like, within a few months before that happened, and this hadn't been—I'd never been hospitalized before, and so I don't know if there was any relationship between that."

**Distressing sensory experiences.** Distressing sensory experiences were identified in a few participant accounts of undesired mental health impacts. These experiences included unpleasant or even scary sensory experiences that impacted participants during their psilocybin use, as well as for varying durations afterward.

In a notably severe case, participant 117-BD1 described a terrifying experience in which she was "certain [she] had died." Clinically significant symptom increases (e.g., mood instability, confusion, and terror) lasted for months and necessitated psychiatric services. Later, she was given a diagnosis of dissociative trauma disorder.

> "I became aware that I was in this realm of nothingness [. . .] it was space itself. It was the deepest, darkest, densest black, like, the gnarliest silence [. . .] I've never experienced anything like it [. . .] a couple seconds later, I realized that I was not—there was no body, there was no more. [. . .] I couldn't remember my name. [. . .] It was a feeling of trying to lift my shoulders, but there was no body. Like, it was gone, and so it was this mourning."

Another participant reported experiencing auditory hallucinations during psilocybin's peak effects that required later medication intervention. Participant 115-BD1 shared:

> "[. . .] And then I'm, like, hearing my husband laughing, and they're talking, and I turn and look at him, and his mouth isn't moving, and I was like, *holy shit*."

Other participants explained that the comedown from psilocybin was most unpleasant. For instance, 120-BD1 called the comedown experience "irritating" and "horrible," stating:

> "Every time I would come down [. . .] I would hallucinate scary things. And it was a feeling of like my skin crawling."

Though less distressing, 122-BD2 shared that he experienced an increase in visual "after imaging" following psilocybin, and some more prominent tinnitus symptoms, which he described as "not anything too life altering":

> "It's just like, like tinnitus ear ringing. And like I've been checked for like tinnitus before because like I've had that, like my ears ringing before. And they said like my ears were perfectly fine. [. . .] But I will say like after I have like tripped a couple times [. . .] it is like a bit more prominent."

## Contextual factors

Contextual factors reported as influential to some participant experiences were (1) polysubstance use and psilocybin dose, (2) solo versus social experiences, and (3) pre-psilocybin sleep deprivation. Among key contextual factors, endorsements of using several consciousness-altering substances before or during psilocybin experiences was often associated with negative impacts, though not exclusively. Psilocybin dosage (e.g., unknown dose or taking too much) was another factor associated with undesired impacts. Variability across contextual factors and outcomes was a common feature in our findings. For example, social context of participant experiences included a range of contexts and outcomes (e.g., environmental stimuli). Sleep deprivation prior to psilocybin use was observed in some participant narratives of undesired impacts.

**Poly-substance use and psilocybin dose.**   Among the key contextual factors reported by participants as relevant to undesired impacts were endorsements of the consumption of several consciousness-altering substances before or simultaneously to their psilocybin experience. While a few participants noted benefits of mixing substances, undesired impacts were more commonly implicated. Psilocybin dosage (e.g., unknown dose or taking too much) was another factor participants referenced.

Providing context of worsening of symptoms, participant 123-BD1 described poly-substance use:

> "I feel like the night that the mania started, I was like drinking, smoking weed, and took a whole bunch of mushrooms. And I feel like that's what tipped me over the edge."

Participant 123-BD1 noted that the combined impact of these substances led to her "letting someone cross personal boundaries, which ended up being traumatic." Participant 117-BD1 also endorsed poly-substance use that she stated may have impacted the traumatic psilocybin experience she later had:

> "I also had smoked marijuana coming up, and I later read that that's maybe not the best idea, so that might have had something to do with [the traumatic psilocybin experience], as well."

In contrast to reports of poly-substance use leading to undesired impacts, other participants reported intentions to experience psilocybin on its own, without other substances. As participant 103-BD1 noted:

> "I've never mixed drugs because it didn't seem like a good idea to me. Like, I didn't even take my medications the day I was going to take the mushrooms because I didn't know what kind of interactions there would be and I wanted to just give myself as clean a slate as possible."

Describing an undesired impact, participant 112-BD1 stated, "I think I may have taken too much. [. . .] Those two bad trips it was—I think the improper dosage and, like, a lack of preparation." Participant 108-BD1 described delayed onset of psychedelic effect as a reason she "took too much" psilocybin-containing chocolate when she experienced purging and, "the walls of the room felt like I was in a casino [. . .] all colors." Elaborating on this, she remarked:

> "I would have thought that of any time that could have initiated a manic experience, that that would have been the time, but there was nothing after it."

**Solo versus social experience.**   Participants described the experience of taking psilocybin alone, compared to taking psilocybin with one, a few, or many other people. Multiple participants seemed to prefer experiencing psilocybin alone due to their ability to attend to their inner experience without distraction. Participant 121-BD1 stated:

> "When I [take psilocybin] by myself it's a bit different. I'm more focused on myself. It's just different thing, but it's as much enjoyable."

Beyond limiting social encounters, participants also stated that limiting external stimuli during psilocybin use can increase self-attunement. Participant 101-BD1 shared:

"If I was to sit at home and focus, and meditate, maybe sit in a closet, a dark room [. . .] and inhibit any kind of audio input or visual, any kind, that'll really allow the substance to take its course."

Participant 112-BD1 echoed concern that using psilocybin in stimulating social environments could lead to unintended outcomes. She reported having her "worst manic episode" after taking psilocybin in a bustling festival setting, stating:

"It was also in a festival environment [when taking psilocybin], so, like, I just was, like, so energetic, so interactive, like, so hyper [. . .] I couldn't slow down [. . .] And I ended up having a manic episode after that."

By contrast, many participants reported taking psilocybin with close others as a way of having fun or enhancing relational bonds. Some of these experiences were held in people's homes, while others occurred outdoors in nature. In sharing an experience of taking psilocybin socially, participant 109-BD1 said:

"I was with friends camping and we all decided that we were going to trip, and it was much more of, like, a social experience. And I would say it was more, like, positive. Like, it felt really nice to [. . .] be outside, and explore nature, and see the stars. And after that experience, the next day, I just felt kind of more, like, this lightness feeling."

Another reason participants reported using psilocybin with others was to enhance safety. Participant 103-BD1 stated she never consumes mushrooms "in public" due to the uncertainty of that environment. In addition to ensuring she is in a private environment when taking psilocybin, participant 103-BD1 stated:

"There's always somebody [. . .] designated as the sober person who was also trained as a medic, or a nurse, or a something just in case something [goes] wrong."

**Pre-psilocybin sleep deprivation.** A pattern of association between pre-psilocybin sleep deprivation and undesired dosing impacts was observed among some participant narratives. Although participants mainly discussed how psilocybin use impacted later sleep quality, a few participants noted ways that impaired sleep prior to psilocybin use impacted their experiences.

Participant 112-BD1 shared that impaired sleep prior to taking psilocybin creates a "perfect storm" of adverse outcomes. In describing an experience of mania following psilocybin use, participant 112-BD1 detailed the context of her psilocybin dosing experience by stating:

"The environment was, like, really loud and, like, you know, flamboyant and fun. And I was also doing a lot of psychedelics and not sleeping very much."

Participant 101-BD1, who endorsed an intention to get "high" multiple times throughout the interview, also described the relationship between sleep deprivation and psilocybin use:

"I like to be sleep deprived and then use hallucinogenics because it would force my body to get super high on hallucinogenics. Only bad thing about that is I feel like it just allows you to be open more to spiritual warfare when you do stuff like that."

## Discussion

The present study aims to understand subjective experiences of individuals with BD who have consumed psilocybin-containing mushrooms to identify possible benefits, risks, as well as contextual factors that affected outcomes after use. To our knowledge, this is the first qualitative study of individuals with BD who have used these mushrooms. We found that during and following full-dose psilocybin use, adults with BD underwent a range of experiences that could generally be categorized as either improving their mental health and functioning or eliciting undesired mental health impacts (although it is important to note that our sampling procedure enriched for individuals who reported positive or negative outcomes after psilocybin use). Noteworthy contextual factors of "set" and "setting" [18, 60] that participants stated may have influenced these beneficial and undesired outcomes included poly-substance use and dose, social context (i.e., psilocybin taken alone or with others), and sleep prior to use. Belser and colleagues [46] highlight the value of qualitative approaches in emergent psychedelic research as *hypothesis generating*, in contrast to *hypothesis testing* in quantitative analysis. Here, thematic analysis [52] provided a flexible approach to explore and conceptualize variation among individuals' experiences in the context of this understudied population in psychedelic science [32, 43]. Despite challenging dosing experiences and undesired impacts, our findings suggest that increased emotion processing may be an important change process related to reductions in depressive symptoms, as well as to enabling formation of helpful new perspectives for individuals with BD. Variability across undesired and beneficial sleep changes warrants further investigation.

BD depression is a tremendous burden for individuals living with this diagnosis [2, 3]. Among the mental health improvements reported by participants after psilocybin use were positive mood changes and reduced severity and frequency of depressive episodes. Such symptom remission occurred during and after their psilocybin experiences, with mood and functional improvements lasting days, weeks, or months. Though some participants endorsed continued depressive symptoms, they reported feeling less debilitated by functional impairment. Change processes in psychedelic therapy remains an important emergent area of research [46, 61]. Individuals living with a diagnosis of BD may experience challenges with emotion regulation and processing as a part of this condition [62]; our findings suggest further investigation in the area of increased emotion processing during psilocybin experiences as a key change process in relation to symptom changes. For example, individuals attributed their decreased depressive symptoms to the emotional release of painful or distressing content from their lives, such as grief or childhood issues, that were otherwise suppressed or avoided. Participants reported developing a deeper awareness of their internal cognitive and emotional realms based on the insights that occurred during psilocybin sessions. Importantly, contacting uncomfortable psychic material was said to generate new perspectives and aid participants in feeling less impacted by day-to-day challenges. Participants described relating to themselves, other people, the world, and their BD symptoms in new and helpful ways. Agin-Liebes and colleagues [61] have suggested that psilocybin-assisted psychotherapy enables freedom from emotionally avoidant tendencies through mindful, experiential modes of processing disowned feelings (e.g., grief). Reductions in emotion avoidance, or what in the transdiagnostic and process-based Acceptance and Commitment Therapy (ACT) model is termed "experiential avoidance," have been observed to precipitate improved well-being by relieving suffering through increased psychological flexibility and present-oriented awareness [63]. Change processes identified through this analysis suggest findings similar to those in which patients with unipolar depression saw a shift from avoidance (of emotion) to acceptance [64]. Our findings suggest further research, both qualitative and mix-methods, in this area for individuals living with BD is warranted.

Participants experienced a multifaceted array of undesired mental health impacts during and following psilocybin use. Qualitative findings here elaborated on the Phase I results by contributing contextual information pertaining to the "set" and "setting" [18, 60] in which psilocybin was used. Unsurprisingly, given the findings by Morton and colleagues [43] in Phase I of the over-arching study, we observed decreases in quantity and quality of sleep following psilocybin use. Sleep disruptions are common to this population [65], and persist across phases of the condition (depressive, manic, euthymic) [66]. In their expert review, Gold and Sylvia [66] present evidence for sleep–wake rhythms as bio markers of BD depression, including shortened rapid eye movement (REM) sleep latency, increased REM sleep density, and shortened slow-wave sleep latency. Such alterations of REM sleep are also prominent features of unipolar depression sleep architecture [67]. Compared to individuals with unipolar depression, a larger number of young people with BD depression experience delayed sleep onset [68], and are more likely to be evening types when depressive symptoms are elevated [69]. For some participants in our study, sleep changes following psilocybin use were associated with adverse outcomes. While some took precautions to prevent poor sleep, interestingly, a few participants reported greater relaxation and easeful sleep the night after or even days following psilocybin use. Experiences of increased relaxation (e.g., feeling calm and grounded) among individuals with BD reported here suggest similar psilocybin effects as reported elsewhere. Psilocybin therapy has been associated with reduced distress, anxiety, and depression [25, 70–72], as well as increased mindfulness [73], which all could have contributed to improved relaxation and sleep in our population.

As sleep is central to mood and emotion regulation, and because prior emotional events can strongly impact sleep quality more generally [74], the bidirectionality of daytime affect regulation and nighttime sleep disturbance is of particular importance for individuals with BD [75]. There are important clinical implications of sleep and abnormalities of circadian functioning in BD [76], and retrospectively self-reported evidence indicated that sleep disturbance is experienced as the most common early indication of mania [77]. Some participants here associated poor sleep with exacerbating symptoms of hypo/mania that often necessitated medication management and even hospitalization. Sleep disturbances prior to psilocybin use were less commonly reported. However, these instances were associated with undesired outcomes, such as when psilocybin was consumed in a stimulating social environment or festival context. Froese and colleagues [78] have highlighted sleep as an underexplored topic in current psychedelic literature and hypothesize a potentially beneficial interaction with the functions of sleep in relation to antidepressant action. In a placebo controlled study of the effect of psilocybin on sleep (N = 20; 10 women, age 28–53), Dudysová and colleagues [79] found prolonged REM sleep latency after psilocybin administration in addition to a trend toward decreased overall REM sleep duration. They also found that psilocybin suppresses slow-wave activity in the first cycle of sleep and concluded that their findings suggest psilocybin's antidepressant potential may relate to changes in sleep architecture [79]. While research on the effect of psychedelics on sleep in humans is limited [78], animals treated with ibogaine showed increases in wakefulness and REM sleep latency, and decreases in slow-wave sleep and REM sleep time [80]. Similarly, psilocin has been shown to alter sleep-wake architecture in lab mice [81]. Considering the integral role of sleep in all phases of BD, as well as in quality of life and treatment outcomes in this population [66], our findings here and in Phase I suggest the importance of further investigation into the role of sleep changes on BD participant outcomes following psilocybin administration.

Our qualitative findings elaborated on the Phase I results by contributing contextual information pertaining to the "set" and "setting" [18, 60] in which psilocybin was used. The most salient pattern we observed across contextual factors and outcomes of use reports by

participants was variability. Sleep disturbances and poly-substance use were most commonly reported proximal to undesired mental health impacts, though not exclusively. Some participants reported taking the same psilocybin dose on more than one occasion with differing outcomes (e.g., increased mania severity one of two times), and of combining psilocybin alongside particular substances (e.g., cannabis, rapé) without reported issues. Cannabis use has been associated with prolonged symptoms in individuals with bipolar spectrum disorders [82], and our findings suggest limitations on cannabis use and other substances in the context of future clinical trials are warranted. Psilocybin dosage was another topic of concern for some participants who reported problems after taking inappropriate amounts (e.g., taking too much) of psilocybin, or consuming psilocybin alongside other substances. As previously noted by Gard and colleagues [32], psilocybin use itself may be an indicator of risk-taking behavior during manic prodrome; contextual factors noted by participants in the present study, such as reduced sleep prior to psilocybin consumption and/or polysubstance use, may be interpreted in line with this theory. While diminished insight [83] and risky behaviors associated with episodes of hypo/mania [84] may have contributed to undesired impacts here, uncontrolled factors inherent to naturalistic settings such as dose, environmental stimuli, and poly-substance use must also be taken into consideration when interpreting these findings. Given variability presented here, further research is clearly required to parse out the dynamics of symptom escalation, substance use, contextual factors, and psilocybin consumption in the context of the controlled setting of a clinical trial to carefully determine risks and safety issues for this population.

Interestingly, some participants who endorsed challenging psilocybin experiences with marked symptom increases paradoxically remarked that these experiences were later found beneficial and perspective shaping. The most pronounced of these accounts was a participant who believed she had died during a solo psilocybin experience and later required psychiatric services due to clinically significant symptom increases (e.g., mood instability, confusion, and terror) that exceeded pre-psilocybin concerns and lasted for months. Despite this, the interviewee reported newfound gratitude and a renewed life perspective following her challenging experience. Belser and colleagues [46] have highlighted the need for alternative hypotheses to bolster current theoretical conceptualizations supporting our understanding of mechanisms of action associated with psilocybin. Related increases in openness [85] and changes in life satisfaction have been observed [86] following psilocybin-assisted psychotherapy. Paradoxical elements in our findings echo domains described in the post-traumatic growth literature. These include: improved relations with others, increased perception of personal strengths, identification of new possibilities for one's life, enhanced appreciation of life, and spiritual growth following a traumatic event [87, 88]. Taken together, our findings suggest that we may begin to view dangerous dosing experiences (e.g., poly-substance use or unsupervised mixing of psychedelics with pharmaceutical medication) as distinct from challenging dosing experiences involving unprocessed memories or other psychic material. Importantly, we might move from questioning *whether* psilocybin can be safely administered in clinical trials to those with BD, to asking *how* adults with BD can safely be included in clinical psilocybin studies.

## Future directions

While these data indicate there are clear risks of adults with BD using psilocybin, there is also evidence that members of this population can use psilocybin safely, and even beneficially. Along with precautions described by Morton and colleagues [43], our findings here suggest that more research in the context of a small feasibility trial is needed to evaluate safety and preliminary efficacy of psilocybin as a treatment for BD depression in a controlled setting. In

order to promote safety for populations with BD in clinical trial design, the following precautions would be advised:

1. Support participants in getting optimal sleep the nights prior to and following dosing; provide psycho-educative guidance on sleep hygiene in the context of trial.

2. Take steps to limit the amount of stimulation in participants' environments during and following dosing.

3. Ensure participants have adequate person-centered therapeutic support from study team during and following dosing as well as designated support person chosen by participant (i.e., friend, family member, partner).

4. Conduct regular and timely assessments of hypo/mania, suicidality, and other risk issues including mixed episode surrounding dosing to monitor for any new or emergent symptoms prior to and after study drug administration.

5. Refrain from dosing individuals who show evidence of mania, hypomania, or a mixed state immediately before dosing.

6. Limit cannabis use and other recreational substances during trial participation.

## Strengths and limitations

Several limitations must be taken into account when reviewing the results of this study. Firstly, although measures were taken to diversify the sample in Phase I [43], the sample, and therefore this subsample, was largely white. Second, because interview participation was limited to people living in North America for logistical ease, the results may not be generalizable to people with BD living in other parts of the world. Third, because participants were self-selected by their willingness to talk to researchers about the consumption of psilocybin, interviews may exclude the perspectives of individuals who felt uncomfortable disclosing details of their experiences for various reasons (e.g., stigma, legal issues, confidentiality, discomfort related to adverse experiences). Next, participants may have answered questions in a way they perceived as helpful or desirable, which may limit transferability. Similarly, given that Phase II explored individuals' prior experiences of psilocybin use in uncontrolled settings, post-event processing and recall bias may have impacted self-report interview responses. Finally, determining the root cause of undesired outcomes is complicated, especially given variability in our findings, and therefore interpretations of these qualitative study results should be considered cautiously. The flexibility of thematic analysis [52, 53] here allowed for conceptualization of important variation among experiences of this understudied population [32, 43], while purposive subsampling [49] supported participant selection to justify sample adequacy [89]. Although our sample here contained a greater proportion of participants with BD-I, individuals who reported negative outcomes in Phase I did not differ from those who did not experience side effects in terms of age, $t(471) = 1.73$, $p = 0.08$, or gender $\chi^2(1, N = 442) = 0.33$, $p = 0.57$. Nor did these groups differ in terms of BD subtype diagnosis [43]. This emergent area of research will no doubt benefit from further scientific investigation.

## Conclusion

This present qualitative study is the second phase of an explanatory sequential mixed methods investigation developed to further understand outcomes of psilocybin use among individuals with BD, with the overarching aim to understand how risk factors and potential benefits could be managed in future clinical trials. Here, we have demonstrated there are clear risks for this

population. We have also provided evidence that adults living with BD have experienced mental health improvements following psilocybin use. Given the long history of excluding this population from psychedelic science, lack of clear evidence for these exclusions, [19, 31, 32] the significant burden of depression for individuals living with a BD diagnosis [5, 90–92], and limited treatment options available [12], more research is urgently needed.

Findings from this study will guide the design of our upcoming clinical trial at the University of California, San Francisco aimed at identifying the safety and feasibility of psilocybin-assisted psychotherapy as a novel treatment for symptoms of depression in adults aged 30–65 living with a diagnosis of BD-II. Our hope is that this limited age range and open-label dose escalation study design will further current understanding of initial safety and efficacy of a psychedelic therapy for individuals with BD experiencing symptoms of depression while also minimizing risk.

## Supporting information

**S1 File. Qualitative interview guide.**
(PDF)

## Acknowledgments

The authors gratefully acknowledge the research participants who were involved in this project, the expertise of the CREST.BD Community Advisory Group, and network members.

## Author Contributions

**Conceptualization:** Mollie Pleet, Emma Morton, Amir Ashtari, Josh Woolley, Erin Michalak.

**Data curation:** Meghan DellaCrosse.

**Formal analysis:** Meghan DellaCrosse, Mollie Pleet, Emma Morton, Erin Michalak.

**Methodology:** Meghan DellaCrosse, Emma Morton, Amir Ashtari, Erin Michalak.

**Project administration:** Mollie Pleet, Kimberly Sakai.

**Supervision:** Josh Woolley, Erin Michalak.

**Writing – original draft:** Meghan DellaCrosse.

**Writing – review & editing:** Meghan DellaCrosse, Mollie Pleet, Emma Morton, Amir Ashtari, Kimberly Sakai, Josh Woolley, Erin Michalak.

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
