## [Decision Letter · Decision Letter 0]

25 Aug 2022

PONE-D-22-16805“A sense of the bigger picture:” A qualitative analysis of follow-up interviews with people with bipolar disorder who self-reported psilocybin use.PLOS ONE

Dear Dr. DellaCrosse,

Thank you for submitting your manuscript to PLOS ONE. After careful consideration, we feel that it has merit but does not fully meet PLOS ONE’s publication criteria as it currently stands. Therefore, we invite you to submit a revised version of the manuscript that addresses the points raised during the review process. Please submit your revised manuscript by Oct 09 2022 11:59PM. If you will need more time than this to complete your revisions, please reply to this message or contact the journal office at plosone@plos.org. Please include the following items when submitting your revised manuscript:A rebuttal letter that responds to each point raised by the academic editor and reviewer(s). You should upload this letter as a separate file labeled 'Response to Reviewers'.A marked-up copy of your manuscript that highlights changes made to the original version. You should upload this as a separate file labeled 'Revised Manuscript with Track Changes'.An unmarked version of your revised paper without tracked changes. You should upload this as a separate file labeled 'Manuscript'.

We look forward to receiving your revised manuscript.

Kind regards,

Sidarta Ribeiro

Academic Editor

PLOS ONE

Journal Requirements:

a) Did participants provide their written or verbal informed consent to participate in this study?

   "EEM has received support for patient educational events from Otsuka-Lundbeck Foundation. JW is currently consulting on scientific protocol development for Alvarius Pharmaceuticals, and previously consulted on protocol development for Psilo Scientific Ltd. and Silo Pharma. This does not in any way alter our adherence to PLOS publishing policies."

5. Your abstract cannot contain citations. Please only include citations in the body text of the manuscript, and ensure that they remain in ascending numerical order on first mention.

Reviewers' comments:

Reviewer's Responses to Questions

**Comments to the Author**

1. Is the manuscript technically sound, and do the data support the conclusions?

Reviewer #1: Yes

Reviewer #2: Partly

2. Has the statistical analysis been performed appropriately and rigorously? 

Reviewer #1: Yes

Reviewer #2: N/A

3. Have the authors made all data underlying the findings in their manuscript fully available?

Reviewer #1: No

Reviewer #2: No

4. Is the manuscript presented in an intelligible fashion and written in standard English?

Reviewer #1: Yes

Reviewer #2: Yes

5. Review Comments to the Author

Reviewer #1: Overall, this is a well-written paper that discusses an important and little-discussed, albeit somewhat contentious, subject and clearly explains the approach utilized.

The manuscript could use some changes, and I'll mention three more relevant problems in the paper at the end of my comments.

The following are comments on the passages, with page and line numbers as they appear in the manuscript.

INTRODUCTION

Page 3, lines 56-57. What do you mean by 'low efficacy'? Please be a bit more precise about the issue.

Pg. 4, ln. 66. To the best of my knowledge, ritual use of psilocybin-containing mushrooms has only been documented with certainty among the Mazatecs of Mexico. Please double-check that the referenced text is neither unduly generic nor 'generous' with ethnobotanical data.

Pg. 5, ln. 87. It is unclear at this point why the evidence that psilocybin actually induces TEAS is not 'convincing'. Only in the next paragraph is this made clear. Please modify this sentence to fit inside the text's argumentative thread.

METHODS

Pg. 9, ln. 193. The text reads 'Audio-recorded Phase II', but previously (pg. 9, ln. 176) it is stated that interviews were 'audio-video recorded'. Please clarify.

RESULTS

Pg. 11, Table 1. Since participant 114 identify themself as gender-queer, why not refer to them as 'Latinx' rather than 'Latino'?

Pg. 14, p. 277. Please briefly define here or in the discussion (pg. 28, p 603) what is emotion processing.

Pg. 16, lns. 319-22. The definition of integration - a commonly used but vaguely defined term in the field of psychedelic research - is unclear, as is the type of integration process that occurred in the case of participant 112.

Pg. 19, lns. 393-4. Participant 115 is described in Table 1 as BD1, but in this passage she describes hallucinations, a psychotic feature. Please clarify if these symptoms were self-limiting and/or why she was not diagnosed with BD with psychotic characteristics.

Now I'll address the two most significant issues.

1. Responses of persons with BD to mushrooms containing psilocybin vary significantly. I missed a table that indicated how each participant was included or excluded from each category/theme. I understand that this type of resource would assist us in better understanding what relationships may exist, for example, between having a challenging experience or not and presenting mood improvement or sleep effects (whether insomnia or improved sleep) afterward - and whether such relationships exist. Please include this table or explain why it should not be included.

2. The 'Future directions' section (pg. 33) appears to be a little premature in offering solutions to the problem of BD interaction with psilocybin consumption, especially for qualitative research that, as the text itself states, should raise questions rather than provide answers. Before outlining how psilocybin may be given to persons with BD in clinical trials, I believe you should specify what types of study could and should be conducted - not to mention the ethical aspects - to address this issue.

3. Finally, data availability was not mentioned in the text, as far as I could tell. Please specify where you plan to publish the interview transcripts or provide a justification—such as ethical concerns—for why you are unable to do so.

Reviewer #2: This study consists of qualitative analysis of interviews with bipolar disorder (BD) patients who engaged in the use of psychedelics. This article has several limitations, and most of them are acknowledged by the authors. I will not criticize the limitations intrinsic to the method, but try to raise concerns that can improve the quality of the manuscript. I leave to the Editor the decision of whether purely qualitative analyses are within the scope of this journal.

1. Throughout the manuscript, I feel that the authors are somewhat ambiguous about the potential risk of psychedelics in BD patients, which in my opinion is odd given the available evidence. Thus, in the introduction they state that “little evidence” is usually given to exclude BD patients from trials. In the next paragraphs the authors question the exclusion of BD patients based on the possibility of TEAS. Yet in the next paragraph a study of a small sample of subjects suggests that BD patients are indeed at risk of developing manic episodes due to psilocybin. Perhaps most surprising is the study by Morton et al. cited below, which states that 541 participants who self-reported BD, 32% (174) reported worsening of their symptoms, including feelings of mania, anxiety and difficulty sleeping. But near the end of the manuscript, the authors state that the present study demonstrated “clear risks” for this population, yet there is a trial underway to test the safety of psilocybin on BD patients. Overall, based on this evidence, my feeling is that 1) there is evidence of adverse effects beyond the anecdotal (Morton et al., 30% adverse effects seems highly significant), 2) there is a plausible candidate mechanism by which these adverse effects manifest (TEAS), 3) there is no discussion concerning how to alleviate these potential risks, or to select a sample with mitigated risk to develop adverse symptoms. In this case, I simply wonder how the risks to the patients will be justified in the upcoming trial.

2. I don’t think the criteria to select the subsample was clearly established. Ensuring diversity and representation of minorities might be not enough to select the sample. At the moment, I feel that the process could have been ambiguous, which is not good practice in a study that dramatically reduces the sample size by selecting a subgroup one order of magnitud less than the original.

3. I couldn’t find info concerning medications. Were patients under mood stabilizers, antidepressants, antipsychotics? Did they inform their current use of medication? If affirmative, what are the potential effects of psychedelics interacting with these medications? Some of them are likely to reduce the intensity of the effects, for instance.

4. Authors wrote that “adults with BD underwent a range of experiences that could generally be categorized as either improving their mental health and functioning or eliciting undesired mental health impacts”. This appears to be circular, since the subgroup was made of subjects who had very good and very bad experiences with the drug.

6. PLOS authors have the option to publish the peer review history of their article (what does this mean?). If published, this will include your full peer review and any attached files.

Reviewer #1: No

Reviewer #2: No

---

## [Author Response · Author response to Decision Letter 0]

29 Sep 2022

Complete responses to editor along with point-by-point responses are included in the rebuttal letter as requested in the decision email.

---

## [Decision Letter · Decision Letter 1]

1 Dec 2022

“A sense of the bigger picture:” A qualitative analysis of follow-up interviews with people with bipolar disorder who self-reported psilocybin use.

PONE-D-22-16805R1

Dear Dr. DellaCrosse,

We’re pleased to inform you that your manuscript has been judged scientifically suitable for publication and will be formally accepted for publication once it meets all outstanding technical requirements.

Kind regards,

Sidarta Ribeiro

Academic Editor

PLOS ONE

Additional Editor Comments (optional):

Reviewers' comments:

Reviewer's Responses to Questions

**Comments to the Author**

1. If the authors have adequately addressed your comments raised in a previous round of review and you feel that this manuscript is now acceptable for publication, you may indicate that here to bypass the “Comments to the Author” section, enter your conflict of interest statement in the “Confidential to Editor” section, and submit your "Accept" recommendation.

Reviewer #2: All comments have been addressed

2. Is the manuscript technically sound, and do the data support the conclusions?

Reviewer #2: Yes

3. Has the statistical analysis been performed appropriately and rigorously? 

Reviewer #2: N/A

4. Have the authors made all data underlying the findings in their manuscript fully available?

Reviewer #2: Yes

5. Is the manuscript presented in an intelligible fashion and written in standard English?

Reviewer #2: Yes

6. Review Comments to the Author

Reviewer #2: All the comments addressed, thank you.

7. PLOS authors have the option to publish the peer review history of their article (what does this mean?). If published, this will include your full peer review and any attached files.

Reviewer #2: No

---

## [Editor Report · Acceptance letter]

5 Dec 2022

PONE-D-22-16805R1 

“A sense of the bigger picture:” A qualitative analysis of follow-up interviews with people with bipolar disorder who self-reported psilocybin use. 

Dear Dr. DellaCrosse:

I'm pleased to inform you that your manuscript has been deemed suitable for publication in PLOS ONE. Congratulations! Your manuscript is now with our production department. 

Kind regards, 

on behalf of

Sidarta Ribeiro 

Academic Editor

PLOS ONE